# Polymorphisms in Lymphotoxin-Alpha as the “Missing Link” in Prognosticating Favourable Response to Omega-3 Supplementation for Dry Eye Disease: A Narrative Review

**DOI:** 10.3390/ijms24044236

**Published:** 2023-02-20

**Authors:** Benjamin Paik, Louis Tong

**Affiliations:** 1Lee Kong Chian School of Medicine, Nanyang Technological University, Singapore 308232, Singapore; 2Department of Cornea and External Eye Disease, Singapore National Eye Center, Singapore 168751, Singapore; 3Ocular Surface Research Group, Singapore Eye Research Institute, Singapore 169856, Singapore; 4Ophthalmology and Visual Sciences Academic Clinical Program, Duke-NUS Medical School, Singapore 169857, Singapore

**Keywords:** omega-3, dry eye disease, ocular surface disease, inflammation, polymorphism, cytokine, genetics

## Abstract

Elements of inflammation are found in almost all chronic ocular surface disease, such as dry eye disease. The chronicity of such inflammatory disease speaks to the dysregulation of innate and adaptive immunity. There has been a rising interest in omega-3 fatty acids to attenuate inflammation. While many cell-based (in vitro) studies verify the anti-inflammatory effects of omega-3, different human trials report discordant outcomes after supplementation. This may be due to underlying inter-individual differences in inflammatory cytokine metabolism (such as tumor necrosis factor alpha (TNF-α)), in which genetic differences might play a role, such as polymorphisms in the lymphotoxin alpha (LT-α) gene. Inherent TNF-α production affects omega-3 response and is also associated with LT-α genotype. Therefore, LT-α genotype might predict omega-3 response. Using the NIH dbSNP, we analyzed the relative frequency of LT-α polymorphisms among various ethnicities, each weighted by the genotype’s probability of positive response. While the probability of response for unknown LT-α genotypes are 50%, there is greater distinction in response rates between various genotypes. Hence, there is value in genetic testing to prognosticate an individual’s response to omega-3.

## 1. Introduction

### 1.1. Pathogenesis of DED Is Inflammatory

Dry eye disease (DED), characterized by tear film dysfunction, is a disease of the ocular surface characterized by a loss of homeostasis of the tear film [1]. DED is multifactorial, with varying components of aqueous deficiency (due to lacrimal gland dysfunction), mucin deficiency (due to goblet cell loss), and lipid deficiency (due to meibomian gland disease) [2,3,4].

Regardless of the initial precipitating factor, the “final common pathway” in the pathogenesis of DED is localized stress on the corneal surface epithelium (due to localized defects in tear film exposing the underlying cornea), which can lead to the release of inflammatory mediators, progressing to globalized inflammation along the ocular surface, making for an “inflammatory soup” of pro-inflammatory cytokines, lipid mediators, and an influx of proinflammatory cells, which in turn cause further ocular surface damage [2,3].

Such processes potentiate chronic inflammation and are driven by the immune response, which involve both the ocular mucosa, and in many cases, a systemic immune response. These processes are modelled as the “immune response arc” [5,6].

### 1.2. Normal Immune Response Arc: Innate and Adaptive Immunity 

In DED, antigens at a tissue site are phagocytosed by antigen-presenting cells (APCs) such as macrophages and dendritic cells, in which the antigen is cleaved and combined with a unique major histocompatibility complex (MHC) class II peptide (HLA-DR, HLA-DP, HLA-DQ) to form a peptide-MHC complex on their cell surface membrane. These APCs then migrate to lymph nodes and use their peptide-MHC complex to bind to specific receptors on naïve B and T cells, activating them to undergo clonal expansion and differentiation to form effector T cells (CD4+ and CD8+ T cells) and effector B cells (antibody-secreting plasma cells). 

CD4+ T helper (Th) cells play important roles in chronic inflammation, with each subset secreting a unique repertoire of cytokines which regulate the immune response and may act synergistically or antagonistically to each other, of which Th1 cells participate in cell-mediated and delayed hypersensitivity reactions via secretion of cytokines such as tumor necrosis factor-alpha (TNF-α) & interferon-gamma (IFN-y) [7]. Th2 cells also potentiate allergic diseases via secreting interleukins (IL), such as IL-4, IL-5, and IL-10 [8]. A new subset of CD4+ T cells, the Th17 cells, secrete inflammatory cytokines IL-17, IL-21, IL-22, and IL-24 and have been implicated in autoimmune diseases like rheumatoid arthritis [9,10]. 

### 1.3. Inflammation in DED Is Due to Dysregulation of Innate and Adaptive Immunity 

In health, any inflammation is usually acute, self-limiting, and self-resolving. However, failure of resolution of existing acute inflammation can cause continuous chronic inflammation [5]. Elements of inflammation are found in many chronic ocular surface disease, such as DED, blepharitis, meibomian gland disease, allergic conjunctivitis, and rosacea, to varying extents [11,12]. 

#### 1.3.1. Innate Immunity

Desiccating stress induces secretion of IL-1, TNF-α, and IL-6 by ocular surface tissues, which promote migration of ocular resident APCs to the regional draining lymph nodes (LNs) [13]. Tear film hyperosmolarity induces hyperosmolar stress on ocular surface epithelial cells, stimulating production of a mitogen-activated protein kinase (MAPK), nuclear factor-kappa B (NF-KB), IL-1a, TNF-α, and matrix metalloproteinase-9 (MMP-9) [14]. Natural Killer (NK) cells also secrete IFN-y, TNF-α, IL-1, and IL-6, which induce MMP production from corneal epithelial cells to cause corneal epithelial barrier dysfunction [15]. They also promote APC maturation through upregulating MHC-II [15]. Under appropriate stimuli, conjunctival epithelial cells and macrophages secrete many cytokines, such as IL-1B, which in turn stimulates the production of TNF-α and other MMPs [15]. 

#### 1.3.2. Adaptive Immunity 

Th1 cells secrete IFN-y, which upregulates the production of IL-9, CXCL10, and CAMs, facilitating recruitment of more inflammatory cells to the ocular surface as well as TNF-α to supplement those from innate immune cells [15,16]. Th17 cells secrete IL-17, which promotes secretion of a variety of vascular endothelial growth factors (VEGF-A, VEGF-C, and VEGFR-2) to cause corneal lymphangiogenesis, which in turn facilitates an ingress of immune cells to the ocular surface and antagonizes Treg activity, hence permitting Th17 and Th1 cells to expand further, migrate to the ocular surface, and cause more epithelial damage [15,17,18,19,20]. 

### 1.4. Inflammation Is Regulated by Genetics and Environment

These aforementioned inflammatory processes are regulated by both genetic and environmental factors and, hence, must be considered as potential therapeutic targets in the treatment strategy for DED. In particular, genetic factors may affect immune response and, hence, autoimmunity, manifesting in the form of autoimmune disease. However, even in the absence of such autoimmune disease, genetic factors play a role in the immune response in dry eye disease. For instance, polymorphisms in IL-10 (anti-inflammatory) lower its mRNA expression and, hence, the risk of chronic gastritis in Asians [21]. Furthermore, polymorphisms in receptors downstream of cytokine signaling pathways are also associated with dysregulated inflammation, such as in the association of *JAK2* polymorphisms with Behcet’s disease, *TKY2* polymorphisms with Crohn’s, systemic lupus erythematosus (SLE), *STAT3* with psoriasis, *STAT4* with rheumatoid arthritis (RA), and *STAT6* with atopy and asthma [22,23,24,25,26,27]. 

## 2. Role of Omega-3 in Dampening Inflammation 

### 2.1. Omega-3 Fatty Acids as a Treatment for DED

Omega-3 fatty acids have been advocated as a treatment for ocular surface inflammatory disease such as DED. Their use as anti-inflammatory compounds dates back to the 18th century when cod liver oil was used in the Manchester Infirmary to treat rheumatism and other chronic inflammatory conditions like tuberculosis [28]. Studies since then have identified the key compounds in cod liver oil responsible for the anti-inflammatory properties, which are also abundant in seafood and some nuts. These compounds include alpha-linoleic acid (ALA), docosahexaenoic acid (DHA), and eicosapentaenoic acid (EPA), together referred to as the omega-3 fatty acids, which are obtained from the diet alongside omega-6 fatty acids (arachidonic acid) [29,30]. Since these cannot be synthesized from other precursors, they are also referred to as essential fatty acids [31]. 

After absorption in the gastrointestinal tract, these fatty acids compete for metabolism by the same intracellular enzymes (cyclooxygenase (COX) and lipoxygenase (LOX)) to form a variety of lipid mediators of inflammation (eicosanoids), which have various effects on modulating innate and adaptive immune cells [32,33,34]. Supplementation with omega-3 saturates COX and LOX enzymes at the expense of omega-6 metabolism, such that omega-3 metabolites (5-series leukotrienes and 3-series prostaglandins), which are less potent inflammatory mediators than omega-6 metabolites (4-series leukotrienes and 2-series prostaglandins), are preferentially produced, hence shifting the eicosanoid profile from that of a pro-inflammatory milieu to a pro-resolving and anti-inflammatory one [5,32,35]. To this end, studies point towards an optimal omega-3-omega-6 ratio of <4:1 to reduce inflammation [36,37]. Recent evidence has elucidated other metabolites exclusively derived from omega-3, such as resolvins, protectins, and maresins, which promote resolution of inflammation through a variety of mechanisms [38,39]. 

Over the years, omega-3 in DED has been the subject of numerous studies and meta-analyses [1,36,40,41,42]. The pooled effect sizes vary widely, prohibiting the establishment of definitive conclusions. This is because not all studies report positive results, including the recent multicentre randomized trial, the Dry Eye Assessment and Management (DREAM) study, which found no added benefit of omega-3 over placebo, and results from the DREAM extension study showed that patients who stopped omega-3 supplementation did not do worse off than their counterparts who continued on omega-3 [43,44]. 

### 2.2. Effect of Omega-3 Fatty Acids on Innate and Adaptive Immunity 

Not only do omega-3 fatty acids lead to a shift in eicosanoid profile, but they also modulate innate—and to a certain extent—adaptive immunity, and these have been reviewed separately [45]. For example, DHA and EPA can blunt macrophage to M1 phenotype (pro-inflammatory) and instead promote polarization [46] to M2 (involved in promoting tissue repair) [47]. They also have effects on reducing neutrophil migration and reduced antigen presentation by APCs. Regarding adaptive immunity, omega-3 fatty acids can also act on T cell lineages to abrogate Th17 differentiation and promote Treg proliferation. 

Furthermore, omega-3 PUFAs may downregulate inflammatory gene expression. For example, cells pre-treated with DHA or EPA can downregulate *TNF-α*, *IL-1B*, and *IL-6* gene expression [48,49,50]. In particular, omega-3 decreases *TNF-α* expression in monocytes and macrophages, which are the predominant synthesizers of TNF-α [48,51]. A few possible mechanisms include the following: omega-3-mediated decrease in thromboxane A2 production (which itself facilitates cytokine production), or omega-3 fatty acids directly binding to peroxisome proliferator-activated receptors (PPARs), hence inhibiting nuclear factor kappa B (otherwise a stimulant of TNF-α synthesis) [52,53]. Synthesis of omega-3 metabolites like protectin D1 have also been shown to inhibit TNF-α production [54]. 

As shown above, the bioactive metabolites of omega-3 fatty acids play crucial roles in modulating innate and adaptive immunity. As shown, the downstream metabolites of omega-3s are dominated by anti-inflammatory oxylipins, which dampen the production of cytokines by the aforementioned immune cells, of which cytokines from the tumor necrosis factor (TNF) superfamily are extensively implicated in both innate and adaptive immunity. 

## 3. Roles of Inflammatory Cytokines TNF-α and LT-α in DED Inflammation

### 3.1. Tumor Necrosis Factor Alpha (TNF-α)

TNF-α is produced in all nucleated cells, including the epithelial cells of the cornea and conjunctiva [55], as well as in macrophages and monocytes. It is elevated in the tears of DED patients [56,57]. TNF-α performs a vast variety of roles in inflammation via the innate immune response through acting on TNF receptors (TNFR) and TNFR1 and TNFR2, which are found on endothelial cells and leukocytes. TNF-α signaling at these receptors leads to macrophage activation, proliferation, and differentiation [58,59,60]. TNF-α-TNFR1 signaling also upregulates intracellular adhesion molecule-1 (ICAM-1) expression on the vascular (lymph and blood) endothelium on the ocular surface in DED patients, and is, therefore, key in DED pathogenesis [56,61]. TNF-α-TNFR2 signaling stimulates expression of E- and P-selectins which adhere neutrophils, monocytes, and memory T cells to the vascular endothelium, facilitating the ingress of leukocytes to the site of inflammation. TNF-α also induces nitric oxide expression in macrophages, causing local vasodilatation and hyperemia [62,63], and mediates ocular surface epithelial dysfunction (apoptosis, increased cell turnover, squamous metaplasia, staining, etc.) [64]. These processes are summarized in Figure 1. Given TNF-α’s many roles in inflammation, it is no surprise that the TNF-α blockade has been effective in treating ocular inflammatory disorders in both mouse models [65,66] and humans [67,68]. 

Apart from bolstering innate immunity, there is emerging evidence that chronic TNF-α stimulation may potentiate adaptive immune responses by promoting the survival of effector T cells at sites of inflammation, as well as suppressing cell lineages which inhibit cytokine production, such as Treg cells [69,70]. 

### 3.2. Lymphotoxin Alpha (LT-α)

Lymphotoxin-alpha (LT-α) is another cytokine of the TNF superfamily. LT-α is structurally similar to TNF-α and also binds to the TNF1 and TNF2 receptors [71]. It is produced by CD4+, Th17, CD8+, NK, and B cells [72]. It is thought to be mainly pro-inflammatory and exerts its effect through two signaling pathways: Firstly, it binds as a LT-α trimer to TNF1 and TNF2 receptors, through which it causes lymphoangiogenesis, upregulates the secretion of chemokines like Chemokine Ligand 5 (CCL5) and monocyte-chemoattractant-protein-1 (mcp-1), and increases the expression of ICAM in endothelial cells [73,74,75]. Secondly, it can compete with LT-β to form a trimeric ligand which binds to the LT-β receptor, activating pathways like NF-KB [76,77]. 

## 4. Role of Genetics in Prognosticating Anti-Inflammatory Effects of Omega-3 

### 4.1. Genetic Basis for Variation in Omega-3 Response 

Much of the differences in response to omega-3 supplementation between individuals might be due to differences in baseline levels of these omega-3 fatty acids. Wilder’s law of initial value states that the level of response must take into account the initial levels of that substance [78]. Levels of these fatty acids in the eye are not routinely measured, if at all, in the ocular surface where the inflammatory soup of DED resides. This is important, because not all patients with DED will have deficiency of omega-3 in the eye. If the baseline levels of omega-3, omega-6, and their downstream eicosanoid metabolites in the eye are unknown and variable, the outcomes of omega-3 supplementation will likewise be unpredictable [79,80]. In the same vein, baseline pro-inflammatory cytokine levels may affect the level of response sufficient to have a significant effect on inflammation [81,82]. 

Genetic factors may also account for the discrepancies in the anti-inflammatory response to omega-3. This is because single nucleotide polymorphisms (SNPs) in gene coding for inflammatory cytokines can influence the severity and extent of inflammation, of which polymorphisms in *TNF-α* and *LT-α* have been the most extensively studied due to their involvement in both innate and adaptive immunity. For instance, polymorphisms in *TNF-α* show an association with inflammatory connective tissue disease like SLE, RA, and ankylosing spondylitis (AS), and other inflammatory disease like Celiac disease [83,84]. Furthermore, polymorphisms in receptors downstream of TNF-α signaling are also associated with dysregulated inflammation, such as RA, Crohn’s disease, AS, and graft-versus-host disease, which alter binding kinetics between TNFR2 and TNF-α and, hence, the frequency of TNF signal transduction [85]. 

### 4.2. Genetic Basis for Variation in Production of TNF-α 

The regulation of TNF-α production is through many epigenetic mechanisms, none of which has a monopoly on TNF-α production. Germane to the discussion is an understanding of the location of the *TNF-α* gene. *TNF-α* is part of an MHC gene cluster located on chromosome 6, adjacent to *LT-α* and *LT-β*; furthermore, *TNF-α* and *LT-α* share the same transcriptional orientation [86]. At the transcriptional level of regulation, polymorphisms in the *TNF-α* promoter region may affect the affinity of RNA polymerase for the promoter and, hence, the frequency of transcription initiation [87,88]. However, evidence supporting the effect of *TNF-α* promoter SNPs on TNF-α production is not robust. For example, a meta-analysis on a variety of patients (healthy, septic, and autoimmune disease) across a variety of *TNF-α* promoter SNPs (−238G/A, −308G/A, and −857C/T) points to the absence of any association between TNF-α promoter SNPs and TNF-α production [89]. 

Intriguingly, polymorphisms in other genes nearby (LT-α) can also affect TNF-α production [71]. This might be due to a linkage disequilibrium between *LT-α* alleles, *TNF-α* alleles, and even other alleles part of the same gene cluster, such as *HLA-DR* and *HLA-DQ*, to form extended haplotypes [89]. In such situations, polymorphisms of *LT-α* are potentially inherited together as part of an extended haplotype with other genetic polymorphisms, which may affect TNF-α secretion. For example, certain *HLA-DR* subtypes have been correlated with an increased cytokine secretion capacity from monocytes [90]. All these can affect TNF-α production.

At the post-translational level of regulation, soluble TNF receptors may sequester TNF and, hence, attenuate its activity [91]. Additionally, the density of membrane-bound TNF receptors will affect the magnitude of TNF signaling [92]. Additionally, the expression of TNF-binding receptors (TNFR1 and TNFR2) can be regulated. Genetic differences may affect transcription rates or post-translational stability or trafficking of the receptor. TNFR1 promoter polymorphisms (e.g., TNFR1-609G/T) may also affect the binding of transcription factors like interferon consensus sequence-binding protein/interferon regulatory factor 8 (ICSBP/IRF-8) to the promoter, altering TNFR1 expression [93]. Similarly, differences in variable tandem repeat sequences (VNTRs) within the TNFR2 gene promoter correlate with increased promoter activity and, hence, TNFR2 expression [94]. At the genomic level, recent studies have shown that chromatin remodeling and higher-order intra-chromosomal interactions can bring TNF promoter into close proximity with distal enhancers, and histone acetylation can loosen DNA packing around histones and increase promoter accessibility for transcription factor binding [86]. 

### 4.3. Polymorphisms to TNF-α Related Genes Might Predict Omega-3 Response 

Part of the anti-inflammatory effect from omega-3 comes from suppression of TNF-α production. However, if inherent TNF-α production is already low to begin with, any attempts to suppress TNF-α might not further benefit. Conversely, patients with inherently high TNF-α production will be more sensitive to the TNF-α lowering effects of omega-3. 

This hypothesis was evaluated for the first time by Grimble et al. in a study of 111 healthy subjects [95]. To do so, they stratified all subjects by their levels of inherent TNF-α production and pooled all data according to tertiles. In line with their hypothesis, when all subjects were supplemented with omega-3, they found that the highest tertile exhibited a fall in TNF-α. The middle tertile did not respond, and the bottom tertile actually suffered an increase in TNF-α. Lending further support to their hypothesis was the finding that the proportion of subjects with TNF-α suppression was the largest in the highest tertile (86%), followed by the middle tertile (43%) and the smallest (22%) in the lowest tertile. This is shown in Table 1. These differences in baseline TNF-α production and, hence, their responsiveness to omega-3 supplementation might reflect intrinsic differences in the regulation of immunity at baseline. 

Next, Grimble et al. sought to investigate if inherent TNF-α production could be associated with TNF-α or LT-α genotypes. These genotypes arise due to polymorphisms in the *TNF-α* (TNF-α-308G and TNF-α-308A, also known as TNF*1 and TNF*2), as well as *Lymphotoxin A* (LT-α+252G and LT-α+252A a.k.a. TNFB*1 and TNFB*2) genes, giving rise to three alleles at each loci [96]. In their analysis, Grimble et al. evaluated the relative proportion of each genotype in each of the three tertiles of inherent TNF-α production. Three important findings were made. Firstly, TNF-α genotype was unrelated to TNF-α production, as the relative proportions of each genotype were consistent across all tertiles. Secondly, homozygosity for the LT-α+252A allele was positively correlated with inherent TNF-α production (41% homozygosity in the highest tertile and 16% homozygosity in the lowest tertile). Thirdly, heterozygosity at the *LT-α* locus was inversely correlated with inherent TNF-α production. Therefore, the *LT-α* genotype (but not the TNF-α genotype) can affect the inherent TNF-α. These findings regarding LT-α are shown in Table 2.

Therefore, since the *LT-α* genotype is associated with baseline TNF-α, and that the baseline TNF-α tertile predicts direction of omega-3 response, we can deduce that the *LT-α* genotype can potentially predict omega-3 response. Based on the above information, the calculated probability of a particular genotype responding favorably to omega-3 supplementation is the frequency of each tertile in that genotype (Table 2) multiplied by their respective proportion of individuals of each tertile responding favorably to omega-3 supplementation (Table 1). For example, the probability of the B1/B1 genotype response is 0.238 × 0.86 + 0.381 × 0.43 + 0.381 × 0.22 = 0.452. These findings are given in Table 3. 

Since the genetic data reported by Grimble is based on a purely Caucasian population. In order to extrapolate these findings to other ethnicities, we have to assume that the frequency of each TNF-α tertile in a particular genotype, as well as the proportion of individuals responding favorably in each tertile, is similar to that of other ethnicities.

Therefore, if the patient’s genotype is known, the probability of response is taken to be the same amongst all ethnicities. However, if the patient’s genotype is unknown, we can calculate the probability of favorable response in a given ethnic group by considering the relative frequencies of each genotype (B1/B1, B1/B2, and B2/B2) in a given ethnic population. To do so, we obtained the allelic frequencies for each polymorphism from the refSNP database for each ethnic population (Caucasian, African, Asian, etc.). We then used the Hardy–Weidenburg Equilibrium to estimate the proportion of each genotype (B1/B1, B1/B2, and B2/B2) in each ethnic population, as described below [97]:Heterozygosity = 2pq
Homozygous B1/B1 = p^2^
Homozygous B2/B2 = q^2^
where frequency of allele B1 = p; frequency of allele B2 = q. 

These probabilities, referred to as P(unknown), are shown in Table 4 for the major ethnic populations. 

As shown, P(unknown) does not vary much between ethnicities and hovers around 50%, making the probability of favorable response to omega-3 about 50%. However, response rates differ greatly between the various genotypes B1/B1, B1/B2 and B2/B2, as shown in Table 3. Therefore, genotyping will provide a more definitive answer as to whether or not a patient will respond. The aforementioned workflow is summarized in Figure 2. 

While genetic testing may be costly, the cost of omega-3 taken as a supplement over a lifetime justifies its cost, since it allows each patient to make an informed decision on the efficacy of omega-3 supplementation. Furthermore, since this is testing for a single SNP, the cost is not expected to be high. 

### 4.4. Accounting for How Polymorphisms in LT-α Alter the Expression and Activity of Cytokines 

While the effect (while discordant) of *TNF-α* promoter polymorphisms on TNF-α production is clear, the mechanisms behind *LT-α* polymorphisms influencing TNF-α production is less well elucidated. One possible mechanism is that due to the extremely close proximity of the *LT-α* and *TNF-α* gene loci, these LT-α polymorphisms may be in linkage disequilibrium with other unidentified susceptibility polymorphisms responsible for the differences in baseline TNF-α [98]. 

Alternatively, polymorphism in *LT-α* might affect the formation of the transcription initiation complex at the *TNF-α* promoter, again, due to their proximity. The *TNF-α* promoter is just around 2 kilobases downstream of the *LT-α* polymorphism in question [99,100]. Therefore, alterations to the stability of the transcription initiation complex will affect the frequency of the transcription initiation of *TNF-α* and, hence, its levels. However, these need to be verified with future rigorous transcriptomics and metabolomics experiments. 

The polymorphisms in *LT-α* may also affect differences in the expression of tear cytokines which drive DED pathogenesis, although the underlying mechanisms are highly complex. On the one hand, *LT-α* polymorphisms correlate with inherent TNF-α levels, which itself increases pro-inflammatory cytokines IL-6, IFN-y, and IL-1b and suppresses immunoregulatory TGF-β and IL-10 on the ocular surface [65,66]. On the other hand, *LT-α* polymorphisms correlate with plasma LT-α levels, which vary widely among DED patients, and, after a certain LT-α threshold (>700 pg/mL), LT-α positively correlates with increases in both pro-inflammatory (TNF-α, IL-12/23) and anti-inflammatory (IL-10, IL-1b, and IL-1Ra) cytokines, as revealed in a study by Chen et al. in 2020 [101]. Evidently, when DED patients are classified based on LT-α levels, the pathomechanisms are different. Further studies are required to correlate the inherent levels of these other cytokines (individually and in combinations) with omega-3 response. 

Intriguingly, the same study by Chen et al. [101] found no significant differences in tear protein markers between the low LT-α DED (<700 pg/mL) and a healthy control group. The controls could have a mix of both high and low inherent levels of LT-α, leading to no difference when compared to the low LT-α or high LT-α groups. It may be arbitrary to divide LT-α into two categories when it is probably a spectrum. It is unlikely that the entire cytokine profile is dependent on one polymorphism; multiple genetic factors are likely at play. The study also did not report the clinical severity of DED in each of the three groups. Further studies are required to justify the threshold of 700 pg/mL to segregate “high” and “low” LT-α and to perform comparisons which are matched for DED severities. This will elucidate if LT-α is an independent predictor of the tear cytokine profile. 

### 4.5. Implications for Testing and Other Potential Testing Targets 

The aforementioned results also emphasize the need for *LT-α* genotyping prior to starting omega-3 in clinical trials. This is so that patients with similar *LT-α* genotype can be analyzed together so that effects of any variables being tested (e.g., concentration of omega-3, type of omega-3, combinations of omega-3 fatty acids given, etc.) are not confounded by genetic factors which also affect the individual’s response to omega-3. Otherwise, the results might be heterogenous with no clear direction of effect, which is the case with the DREAM study and the DREAM extension study. 

The levels of lipid mediators of inflammation, such as prostaglandin E2 (PGE_2_), may also be measured in the tears of DED patients. This is because high levels of PGE2 suggest an underlying imbalance of pro-inflammatory lipids, which may be counterbalanced by omega-3 supplementation. This can be achieved using mass spectrometry as described previously [102]. 

The polymorphisms in *LT-α* may also affect the results of certain diagnostic tests for DED. For instance, although Schirmer’s I test has been used to confirm severe aqueous-deficient DED, its variability and invasiveness precludes its utility as a routine diagnostic test of tear volume, in particular for evaporative DED where subtle reduction in basal tear volume will be masked by reflex tearing. It also has poor sensitivity in identifying groups of patients with DED [103,104]. This is because *LT-α* polymorphisms (TNFB*2 allele) not only increase inherent TNF-α levels [103] but also increase inherent LT-α levels [105,106], both of which are strongly correlated with increased IL-17A, which in turn correlates with reduced tear production in the Schirmer’s I test [101]. This shows that LT-α polymorphisms may affect Schirmer’s test results. That being said, to the extent that symptoms and signs of DED reflect the underlying ocular surface inflammation (e.g., mediated by TNF-α), the conclusion that certain LT-α polymorphisms favor omega-3 supplementation should still be valid.

Current meta-analyses are plagued by data heterogeneity due to discordant results from different trials, such that there is no clear direction of effect. For instance, the latest meta-analysis in 2022 of eight studies found no improvement to corneal staining, tear breakup time, or Schirmer’s test due to substantial statistical heterogeneity throughout all outcome measures [42], leading to poor quality evidence and perpetually inconclusive results. Judging from clear effect directions in individual patients and some individual studies, it may simply be due to differences in underlying pathophysiology and cytokine profiles among DED patients [4], whether they involve or are independent of TNF-α. When results are adjusted for and stratified according to genotype, then clearer directions of effect may emerge. As elucidated in our review, LT-α and TNF-α are good places to start. In an era of increasing screen use when remote work arrangements become more popular, DED incidence is likely to rise and so will the number of patients that can benefit from omega-3 supplementation, hence making pre-treatment genotyping more important than ever.

### 4.6. Implications for Specific Patient Populations

Cytokines in patients with systemic diseases with DED manifestation may also influence the response to omega-3, although no studies have been in these specific patient populations. For example, Sjogren’s DED patients have higher inherent TNF-α levels than their non-Sjogren’s DED counterparts [107], which may predict more favorable omega-3 response. However, since the levels of TNF-α production are also affected by cytokines, such as IL-10, IL-4, TGF-β, IL-1, IL-2, and IFN-y [108,109,110], diseases with elevations to these cytokines can affect inherent TNF-α production and, hence, omega-3 response. For example, schizophrenia patients have elevated IL-1 relative to healthy controls [111]. However, while these remain speculative, they suggest targets for future research.

## 5. Conclusions

In conclusion, the polymorphisms in LT-α can prognosticate the direction of response to omega-3 supplementation as proxied by changes in TNF-α production. Greater inherent levels of TNF-α production is predictive for favorable response to omega-3 supplementation. While the calculated probability of favorable and therapeutic response to omega-3 ranges around 50% for all ethnicities, there is greater distinction in response rates between various genotypes. This emphasizes the need for LT-α genotyping before starting omega-3 in both long-term treatment and in clinical trials. There may be other cytokines which act on or independently of TNF-α to affect the response to omega-3 supplementation and merit further investigation.

## Figures and Tables

**Figure 1 ijms-24-04236-f001:**
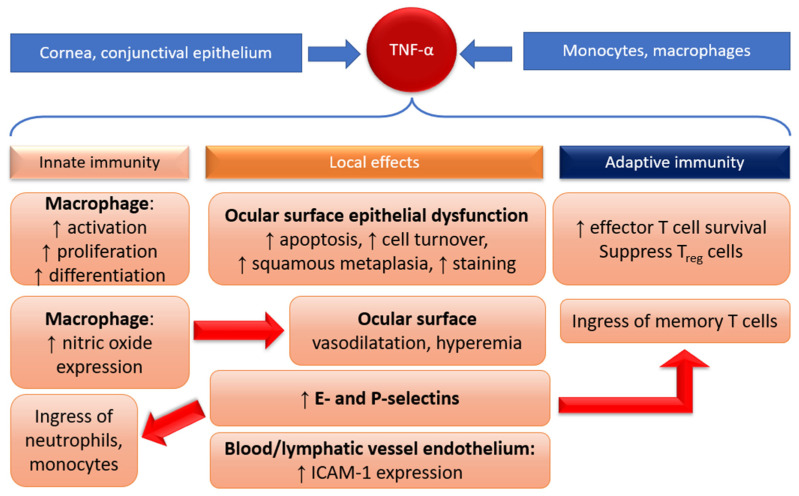
Many roles of TNF-α in potentiating DED inflammation (*ICAM = intracellular adhesion molecule*).

**Figure 2 ijms-24-04236-f002:**
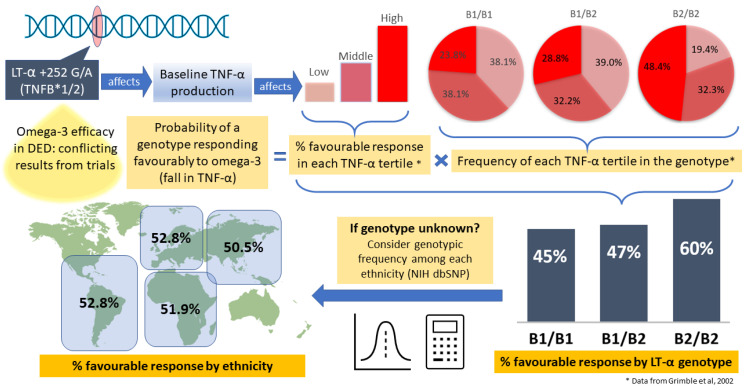
Polymorphisms in lymphotoxin-alpha may prognosticate response to omega-3 supplementation. Data marked with an * is reproduced from Grimble et al., 2002 [95].

**Table 1 ijms-24-04236-t001:** Response of TNF-α production levels by various tertiles of inherent baseline TNF-α production.

		TNF-α Production		
		Before Supplementation (ng/L)	After Supplementation (ng/L)	
	All Subjects (n = 111)	4821 ± 4177	4643 ± 3388	Subjects with Reduction of TNF-α
Tertile of inherent (baseline) TNF-α production	Lowest	1458 ± 600	3809 ± 2571	22%
	Middle	3728 ± 936	4796 ± 3270	43%
	Highest	9277 ± 4338	5323 ± 3941	86%

**Table 2 ijms-24-04236-t002:** Proportion of genotype in each tertile of inherent (baseline) TNF-α production.

		LT-α Genotype		
		B1/B1	B1/B2	B2/B2
	All Subjects (n)	21	59	31
Tertile of inherent (baseline) TNF-α production	Lowest	8 **0.381**	23 **0.390**	6 **0.194**
	Middle	8 **0.381**	19 **0.322**	10 **0.323**
	Highest	5 **0.238**	17 **0.288**	15 **0.484**

Note: bolded figures represent frequency of each tertile in the genotype.

**Table 3 ijms-24-04236-t003:** Probability of response to omega-3 supplementation in each genotype.

Caucasian	Prob in Tertile × Chance of Responding	Probability of Response to Omega-3
B1/B1	0.238 × 0.86 + 0.381 × 0.43 + 0.381 × 0.22	=0.452
B1/B2	0.288 × 0.86 + 0.322 × 0.43 + 0.39 × 0.22	=0.472
B2/B2	0.484 × 0.86 + 0.323 × 0.43 + 0.194 × 0.22	=0.598

**Table 4 ijms-24-04236-t004:** Response to omega-3 supplementation in each ethnicity.

	Proportion B1/B1	P(B1/B1)	Proportion B1/B2	P(B1/B2)	Proportion B2/B2	P(B2/B2)	P(unknown)
European	0.102	0.452	0.435	0.472	0.463	0.598	0.528
African	0.142	0.452	0.470	0.472	0.389	0.598	0.519
Asian	0.204	0.452	0.495	0.472	0.300	0.598	0.505
Latin America	0.102	0.452	0.434	0.472	0.464	0.598	0.528

## Data Availability

The data presented in this study may be available on request from the correspondence author.

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
