# Peer review of "Polymorphisms in Lymphotoxin-Alpha as the “Missing Link” in Prognosticating Favourable Response to Omega-3 Supplementation for Dry Eye Disease: A Narrative Review"

_ijms, 2023, doi:10.3390/ijms24044236_

Round 1

Reviewer 1 Report

This is an interesting review and the authors have collected a dataset using cutting edge methodology. The paper is generally well written and structured. However, the paper has some shortcomings in regards to some data presentation and text, which prevents its publication in the current form. That said, there are several areas in the manuscript that deserve improvements. Below I have provided some remarks on the text:

1.    Although the Schirmer's test has been available for more than a century, several clinical studies have demonstrated the weakness in identifying a large group of patients with dry eyes. While newer and better tests of tear production and function are becoming available, please discuss whether polymorphisms in lymphotoxin-alpha has any roles in affecting the results of these test.

2.    With different diagnostic results, does the conclusion that polymorphisms in lymphotoxin-alpha favors the response of omega-3 supplementation for dry eye disease remains the same?

3.    Please spell the full name of “LN” at first mention in section 1.3.1, though I understand that might be lymph node.

4.    Please discuss whether the polymorphisms in lymphotoxin-alpha affect differences in the expression of tear proteins.

5.    Please discuss whether cytokines in subjects with systemic diseases that might affect DE, such as patients with SS, psychiatric diseases, and malignant tumors, have different effects on favorable response to omega-3 supplementation between normal subjects and patients with DE with high and low LT-a levels.

6.    Some studies suggest no significant differences in tear protein markers between the low LT-a DE group and the control group. Does polymorphisms in lymphotoxin-alpha modulate the effect? Please discuss.

7.    A graphical abstract is a good presentation of your work and appeals to readership. I strongly encourage the authors to make it.

Reviewer 2 Report

The manuscript entitled “Polymorphisms in lymphotoxin-alpha as the missing link in prognosticating favourable response to omega-3 supplementation for dry eye disease: A narrative review is a narrative review on TNF-a production and omega-3 response in patients with dry eye disease. The current review clearly links the polymorphisms in LT-a gene with the differential response to omega-3 supplementation. Overall the manuscript is well written and of high interest to general public. This review would also drive an attention about the prognostic role of genetic polymorphisms in dry eye disease.

However, some of the major comments need to be answered before publication.

1.      Graphical abstract must be provided

2.      Figures must be provided for the better understanding

3.      There are few reviews published recently in the same topic “omega-3 supplementation and dry eye disease”. Therefore, the importance of the current work must be explained.

4.      Full forms must be provided for all the abbreviations when used first time in the manuscript even though they are well-known cytokines like TNF-a, IFNg etc.

5.      Conclusion is too short

Round 2

Reviewer 1 Report

The manuscript is much improved. It is acceptable for publication now.